# Longitudinal Plasma Lipidomics Reveals Distinct Signatures Following Surgery in Patients with Glioblastoma

**DOI:** 10.3390/metabo15100673

**Published:** 2025-10-15

**Authors:** John Paul Aboubechara, Yin Liu, Oliver Fiehn, Lina A. Dahabiyeh, Ruben Fragoso, Han Sung Lee, Jonathan W. Riess, Rawad Hodeify, Orin Bloch, Orwa Aboud

**Affiliations:** 1Department of Neurology, University of California (Davis), Sacramento, CA 95817, USA; jjaboube@health.ucdavis.edu; 2Comprehensive Cancer Center, University of California (Davis), Sacramento, CA 95817, USA; 3Department of Ophthalmology, University of California (Davis), Sacramento, CA 95817, USA; 4West Coast Metabolomics Center, Department of Molecular and Cellular Biology, University of California (Davis), Davis, CA 95616, USA; ofiehn@ucdavis.edu; 5Department of Pharmaceutical Sciences, School of Pharmacy, The University of Jordan, Amman 11942, Jordan; 6Department of Radiation Oncology, University of California (Davis), Sacramento, CA 95817, USA; rcfragoso@health.ucdavis.edu; 7Department of Pathology, University of California (Davis), Sacramento, CA 95817, USA; hnslee@health.ucdavis.edu; 8Department of Internal Medicine, Division of Hematology and Oncology, University of California (Davis), Sacramento, CA 95817, USA; jwriess@health.ucdavis.edu; 9Department of Biotechnology, School of Arts and Sciences, American University of Ras Al-Khaimah, Ras Al Khaimah 72603, United Arab Emirates; 10Department of Neurological Surgery, University of California (Davis), Sacramento, CA 95817, USA

**Keywords:** plasma lipidomics, glioblastoma, liquid biopsy, tumor metabolism, machine learning

## Abstract

Background: Glioblastoma is an aggressive brain tumor that invariably recurs despite treatment, partly due to metabolic adaptations, including altered lipid metabolism. This study investigates plasma lipidomic profiles in patients with glioblastoma to explore their potential as a liquid biopsy for disease monitoring. Methods: Plasma samples were collected from 36 patients with histopathologically confirmed IDH wild-type glioblastoma at four treatment stages: pre-surgery (*n* = 36), post-surgery (*n* = 32), pre-radiation (*n* = 28), and post-radiation (*n* = 17). Untargeted lipidomics analysis was performed using liquid chromatography-high resolution mass spectrometry (LC-HR-MS/MS). Results: Plasma lipidomic signatures differed significantly across treatment stages. Specifically, the lipidomic profile prior to surgery was statistically distinct from those at subsequent stages, demonstrating an increased compound abundance of numerous lipids that are decreased at subsequent stages, including linoleic acid (fold-change 2.58, *p* = 4.21 × 10^−11^), behenic acid (fold-change 2.09, *p* = 9.3 × 10^−10^), and linolenic acid (fold-change 4.44, *p* = 5.83 × 10^−6^). Random forest modeling could predict pre-surgical samples with 85.7% accuracy. Conclusions: Plasma lipidomics shows promise as a potential liquid biopsy approach for monitoring glioblastoma treatment but future studies will need to examine these findings in larger and well-controlled cohorts.

## 1. Introduction

Glioblastoma is an aggressive and lethal primary brain tumor. Despite aggressive treatment strategies involving surgery, radiation, and chemotherapy, glioblastoma invariably recurs [1]. A key factor contributing to recurrence is the ability of surviving tumor cells to acquire new mutations that allow them to adapt and proliferate, which often involves alterations in cellular metabolism [2]. Examination of the metabolic changes occurring within tumors is an area of avid interest; however, little is known about the metabolic changes experienced systemically in patients with glioblastoma. Small retrospective studies have demonstrated an association between metabolic risk factors and survival outcomes [3,4,5,6], which suggests that gaining a deeper understanding of systemic metabolism may grant insights for patient management and potentially improved outcomes.

Clinical management of patients with glioblastoma is hindered by diagnostic limitations of current approaches. Neuroimaging with magnetic resonance imaging (MRI) is the primary modality used to track the tumors’ response to treatment [1]. However, MRI is often unable to distinguish true tumor progression from pseudoprogression—reactive changes that are due to radiation-associated necrosis [1]. As such, there is intense research interest in the development of alternative biomarkers to assess tumor progression [7]. A main approach has been to develop “liquid biopsies” that evaluate the cerebrospinal fluid or blood of patients to identify molecular biomarkers of the tumor [7]. Another approach is evaluating cell-free DNA released by cancer cells [8]. A few promising studies have examined blood plasma metabolomics [9,10,11,12,13,14,15,16]. However, a comprehensive evaluation of plasma lipidomics is lacking. Examination of the plasma lipid profile may hold promise, as glioblastoma has been found to be lipid rich [17], and these lipids may leak across the blood–brain barrier and be detected in the blood [18].

Here, we utilize plasma lipidomics to evaluate the signature of lipids in the blood of patients with glioblastoma with hopes of both developing a sensitive liquid biopsy approach and gleaning biologic insights into systemic lipid metabolism in these patients. We also utilize a machine learning model to enhance the predictive accuracy of this diagnostic approach.

## 2. Materials and Methods

### 2.1. Patient Selection and Sample Collection

This study included 36 patients with IDH wild-type glioblastoma WHO grade 4, confirmed with both histopathologic and molecular testing. This project was approved by the Institutional Review Board of The University of California Davis in 2019 with protocol # UCD 1412052. All patients underwent standard of care treatment with the Stupp protocol [19]. This study specifically restricted enrollment to patients with a new diagnosis of glioblastoma who were candidates for surgery and chemoradiation. Blood samples were collected at four treatment stage time points in order to compare the lipidomic profiles of patients with glioblastoma at different treatment stages: pre-surgery, post-surgery, pre-radiation, and post-radiation. Pre-surgery samples were collected prior to the surgery on the same day during a fasting state. Post-surgery samples were collected two days after surgery. Pre-radiation samples were collected prior to starting the 6 weeks of radiation, which was approximately 2 weeks following surgery due to practicalities of preparing customized radiation plans. Post-radiation samples were collected at the completion of the 6-week radiation cycle. Demographics of the patients are depicted in Table 1 below; some samples were unable to be collected due to practical patient care issues, and those that were collected are designated by an “X” in the table. The samples were spun down and the plasma component was saved for future lipidomic analysis.

### 2.2. Sample Preparation and Lipidomics Analysis

Untargeted plasma lipidomics was performed using liquid chromatography-high resolution mass spectrometry (LC-HR-MS/MS). Plasma or serum samples were thawed in in the refrigerator at 4 °C, and tubes were inverted and then vortexed at low speed for full homogenization. Then, 30 µL sample aliquots were decanted into 1.5 mL Eppendorf tubes and kept on ice. A total of 975 µL of a −20 °C cold solvent mixture of 3:10 (*v*/*v*) MeOH/MTBE (with internal standards) was used to precipitate proteins by vortexing for 10 s, then shaken for 5 min at 4 °C on an orbital mixer. Subsequently, 188 µL LC/MS-grade water was added to each tube at room temperature to induce phase separation. Tubes were vortexed for 20 s and then centrifuged for 2 min at 12,000× *g*. The upper organic phase was transferred to two separate tubes (350 µL/each tube) for lipidomics analysis. Both aliquots were dried down in a centrifuge and kept under nitrogen gas prior to LC-MS/MS analyses at −20 °C to avoid metabolite oxidation during storage or shipping. For quality control, method blanks (30 µL of H_2_O) and pooled plasma or serum controls (30 µL) were extracted with the same procedure along with the actual samples. One blank and one pool QC were used for each set of 10 test samples.

Prior to LC-MS/MS analyses, dried samples were resuspended in 110 µL methanol/toluene (9:1, *v*/*v*). Lipids were separated on a Waters BEH C18 UPLC column (50 × 2.1 mm; 1.7 µm) at 65 °C at a flow-rate of 0.6 mL/min using an Agilent 1290 UPLC, using a gradient from 60:40 (*v*/*v*) acetonitrile/water to 90:10 (*v*/*v*) isopropanol/acetonitrile. The duration of the chromatographic run was 5 min. We used both positive and negative electrospray ionization mass spectrometry on Agilent QTOF mass spectrometers, acquiring MS/MS data at 4 Hz from *m*/*z* 50–1700 at 15,000 resolving power (FWHM), using a capillary voltage of +/− 3 kV and nitrogen gas temperature of 325 °C at 8 L/min. Positive electrospray MS/MS runs were acquired on an Agilent 6530 QTOF mass spectrometer by injecting 1.7 µL per sample, and negative electrospray MS/MS data were acquired on an Agilent 6550 TOF mass spectrometer by injecting 5 µL per sample. To increase the total number of MS/MS spectra, five consecutive runs were acquired for both positive and negative electrospray conditions on pooled QC samples.

All spectra were stored in centroid format. Deconvolution, peak picking, alignment, and compound identification were completed through open source software MS-DIAL v4.8, supported by the largest public MS/MS lipidomics library available, which is included within MS-DIAL [20]. Lipids were identified by accurate mass, MS/MS, and retention time matching including Retip predictions. Recursive data analysis was used to search and replace missing values from raw data files. Annotated lipids were enriched by the following metadata: names, InChI keys, m/z and retention index values, and adduct types. Relative quantification was performed by peak height determination using the precursor ion intensities. Data were normalized to the sum of 76 internal standards (67 internal standards from the SPLASH One Avanti Polar Lipids kit, plus 9 additional standards for free fatty acids and acylcarnitines). SERRF batch correction was not necessary because this was a small set of only 145 samples—SERRF corrections are typically done for larger sets of 300 samples or more [21].

### 2.3. Statistical Analysis

MetaboAnalyst 6.0 was utilized for the metabolomics workflow. For volcano plots, we employed a fold-change threshold of greater than two and a *p*-value threshold based on a false discovery rate of less than 0.05. Bar graphs displaying the mean and standard error of the mean are presented for the most significantly altered lipids; these were generated using GraphPad Prism version 10. Repeated measures analysis of variance (ANOVA) was performed with Tukey post-hoc test for multiple comparisons, and statistical significance was determined as follows: * *p* < 0.05; ** *p* < 0.01, *** *p* < 0.001, and **** *p* < 0.0001. Partial least squares discriminant analysis (PLS-DA) was performed on the entire dataset. The validity and robustness of the PLS-DA model were confirmed using permutation testing and 5-fold cross-validation (Appendix A). Graphs of the first and second principal components were generated to visualize lipidomics signature relationships between treatment time points. Heatmaps were used to visualize the variance of the 25 lipids with the highest F-statistics based on one-way ANOVA. Variable importance in projection (VIP) scores were determined for the top 15 lipids that explain most of the variance of the datasets.

### 2.4. Machine Learning

A random forest model was used for machine learning examination of the lipidomic profiles of different treatment stages. MetaboAnalyst 6.0 was used to perform the analysis, with 500 trees guiding the group discrimination. Out-of-bag error was reported to assess the model’s accuracy in predicting treatment stages.

## 3. Results

### 3.1. Plasma Lipidomic Signatures Differ by Treatment Stage

Examination of the plasma lipidomic profiles of patients with glioblastoma across the four treatment stages demonstrated distinct lipidomic signatures, particularly in the pre-surgery samples. Hierarchical clustering was performed to evaluate the top 25 lipids that best represented the variance of the broader dataset. This demonstrated increased levels of numerous lipids prior to surgery that are reduced at subsequent treatment stages (Figure 1A). The variable importance in projection (VIP) scores of the top 15 lipids that best explain the group differences are depicted in Figure 1B. Partial least squares discriminant analysis (PLS-DA) was performed to examine the variance of the lipids across the entire dataset from all treatment stages. The resulting score plot shows a distinct clustering of the pre-surgery samples in relation to the other treatment stages (Figure 1C). This separation was found to be statistically significant and robust, as confirmed by permutation testing (*p* < 0.001) and five-fold cross-validation (Appendix A).

### 3.2. Surgery Alters the Plasma Lipidome

The lipidomic profile was, next, specifically compared between the pre-surgery and post-surgery treatment stages. Hierarchical clustering demonstrates distinct clustering of the pre-surgery samples from the post-surgery samples, when evaluating the variance driven by the top 25 lipids (Figure 2A). Numerous lipids were found to have increased abundance pre-surgery compared to post-surgery, while only one lipid (CE 22:6) was found to have lower abundance prior to surgery (Table 2). The volcano plot depicts the significant lipids with the fold-change threshold drawn at 2 and the *p*-value threshold being less than 0.1 (Figure 2B). PLS-DA again demonstrates a significant separation between the pre-surgery and post-surgery samples (Figure 2C). This separation is statistically robust, as confirmed by permutation testing (*p* < 0.001), and a two-component model was found to be optimal via cross-validation (Appendix A). VIP scores of the top 15 lipids that best explain the group differences are depicted, with linoleic acid having the largest score (Figure 2D).

### 3.3. Chemoradiation Is Not Associated with Significant Plasma Lipidomic Changes

Concurrent chemoradiation was not associated with any significant alterations in the plasma lipidomic signature. In contrast to the changes seen with surgery, concurrent chemoradiation was not associated with robust alterations in the plasma lipidomic signature. Hierarchical clustering and PLS-DA show some separation between the pre- and post-radiation groups, but the effect is modest (Figure 3A,C). While a permutation test on the PLS-DA model indicates that the separation is statistically significant (*p* = 0.019), the model’s predictive ability as determined by cross-validation is very low (Q^2^ ≈ 0.1), which suggests that it is not a robust classifier (Appendix A). This is consistent with the volcano plot analysis, where no individual lipids met the significance threshold after false-discovery rate correction (Figure 3B). VIP scores of the 15 lipids that best explained group differences are depicted in Figure 3D.

An important observation is that the lipids that were found to have decreased abundance following surgery remained at a low abundance during the later treatment stages of pre-radiation and post-radiation. Three representative lipids that had significant decreases following surgery are quantified in Figure 4, including linoleic acid, behenic acid, and linolenic acid. CE 22:6 is also depicted, as it is the only lipid that increased following surgery (Figure 4).

### 3.4. Random Forest Model Enables Predictive Sample Classification

The random forest machine learning model was employed to examine whether the lipidomic signatures generated from each treatment stage could be employed as predictive tools that could aid in diagnostics. The pre-surgery samples could be identified with a low class error rate of 0.143, which corresponds to an accuracy of 85.7% (Figure 5).

## 4. Discussion

The results presented in this study demonstrate that plasma lipidomic profiles are significantly altered across different treatment stages in our cohort of patients with glioblastoma. The most statistically significant change was seen following surgery, wherein numerous lipids were found to have higher abundance in the pre-surgery samples compared to the post-surgery samples. In fact, as shown in Figure 1C, the pre-surgery samples appear to have a statistically unique signature compared to the other three treatment stages. Additionally, the results of the random forest classification demonstrate that the pre-surgery samples can be predicted with an accuracy of 85.7%.

There are two main novel implications of this work. The first promising aspect is that these results demonstrate that lipidomics has merit as a diagnostic tool that can be employed as a liquid biopsy for patients with glioblastoma. The results of this study demonstrate that we can accurately predict patient groups based on their lipidomic signatures. There are multiple limitations of this work, however, that will need to be addressed in future studies. First, as this was a pilot study, the sample size was limited, so future studies will employ much larger cohorts of patients. These findings are limited to changes seen within patients with glioblastoma at different treatment stages; healthy controls are needed to examine whether this approach holds merit as a diagnostic method for tumor presence. Dietary changes will also need to be accounted for in future studies, as well as fasting status (pre-surgery samples are collected in the fasting state on the morning prior to surgery), as this may be an important confounding variable for our findings and is a current limitation. Lastly, the random forest classification approach suffers from an overfitting bias, which limits its generalizability. However, the results of this work can be used to test future datasets.

The second novel finding may allow for speculation regarding the tumor’s biology. The significant changes observed in lipids like linoleic acid, behenic acid, linolenic acid, and CE 22:6 following surgery suggest a metabolic connection between the tumor and the systemic lipid profile. The results here demonstrate a broad elevation in numerous lipids in the plasma of patients with glioblastoma prior to surgery, including triglycerides (TGs), phosphatidylethanolamines (PEs), phosphatidylcholines (PCs), lysophosphatidylethanolamines (LPEs), free fatty acids (FAs), and diglycerides (DGs). These broad increases may result from leakage of lipids from the tumor, given that these tumors have impaired fatty acid oxidation, which results in lipid accumulation [17]. On the other hand, this lipid accumulation may represent systemic metabolic dysregulation, which has been found in patients with glioblastoma [3]. Elevated levels of the lipids linoleic acid and linolenic acid prior to surgery may be associated with eicosanoids and inflammatory mediators that are suspected of being overexpressed in gliomas [22]. The subsequent sharp decrease following surgery is consistent with the removal of the tumor as a major source of these shed lipids or a resultant shift in the systemic metabolic environment. Elevated levels of triglycerides prior to surgery may reflect the increased energetic demands of the tumor, as well as increased demand for cell membrane components. Interestingly there was a lower level of docosahexaenoate (CE 22:6) in the plasma prior to surgery. CE 22:6 is a critical lipid known for its roles in maintaining neuronal functions [23]. The increase in plasma CE 22:6 following tumor resection may therefore reflect a partial relief of this depletion or sequestration, or a change in the host–tumor metabolic interaction. We also recognize that a more detailed analysis of general lipidome features, such as chain length and desaturation, is needed to fully characterize the observed changes. Such in-depth lipidomic feature analysis will be a key component of our future, larger-scale correlative studies between plasma and tumor tissue lipid profiles.

Future studies will explore the biological relevance of these results by performing correlative analyses with the lipidomic profiles of the same patient’s tumor itself. We also plan to examine healthy controls for comparison, as well as examining the lipidomic profile of patients at the time of tumor recurrence—increased lipid abundance at that time would be expected. If we can demonstrate that this plasma lipidomic signature is indeed originating from lipids shed by the tumor, this may allow for indirect examination of glioblastoma metabolism, particularly in response to treatment and experimental therapeutics.

## 5. Conclusions

Plasma lipidomic profiles are dynamically altered throughout the treatment course of patients with glioblastoma, with the pre-surgical state exhibiting a particularly distinct signature characterized by elevated levels of numerous lipids. The ability of random forest modeling to accurately classify pre-surgical samples underscores the potential of plasma lipidomics as a non-invasive liquid biopsy tool for disease monitoring and potentially early detection. While future studies with larger, multi-center cohorts and the inclusion of healthy controls are essential to validate these findings and address limitations such as potential overfitting, the observed changes in circulating lipids following surgical resection suggest a potential link between the tumor and systemic lipid metabolism. Further investigation into the specific lipids identified and their correlation with tumor tissue lipid profiles holds promise for elucidating the metabolic interplay between the tumor and the host, potentially revealing novel biomarkers for disease progression and therapeutic response in this challenging malignancy.

## Figures and Tables

**Figure 1 metabolites-15-00673-f001:**
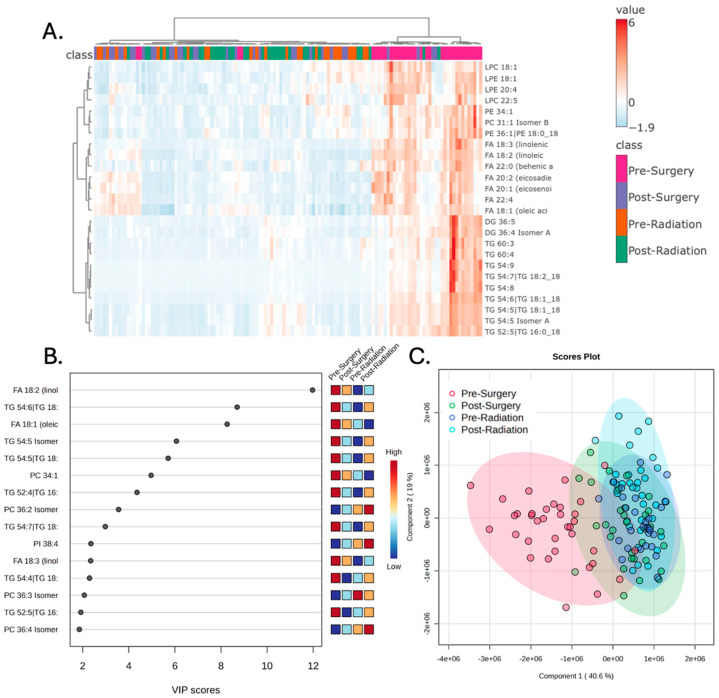
The peripheral lipidomic signature of patients with glioblastoma prior to surgery is distinct from subsequent treatment stages. (**A**) Hierarchical clustering of patient samples based on the 25 lipid metabolites with the highest F-statistic values from a one-way ANOVA. These lipids represent the greatest variance across the treatment stages and demonstrate that pre-surgery samples have a distinct lipidomic profile. (**B**) Variable importance in projection (VIP) scores of the 15 lipids that hold the largest contribution to the dataset’s variance. (**C**) Partial least square discriminant analysis (PLS-DA) demonstrates that the pre-Ssurgery samples separate from the other treatment stages when the first and second principal components are examined.

**Figure 2 metabolites-15-00673-f002:**
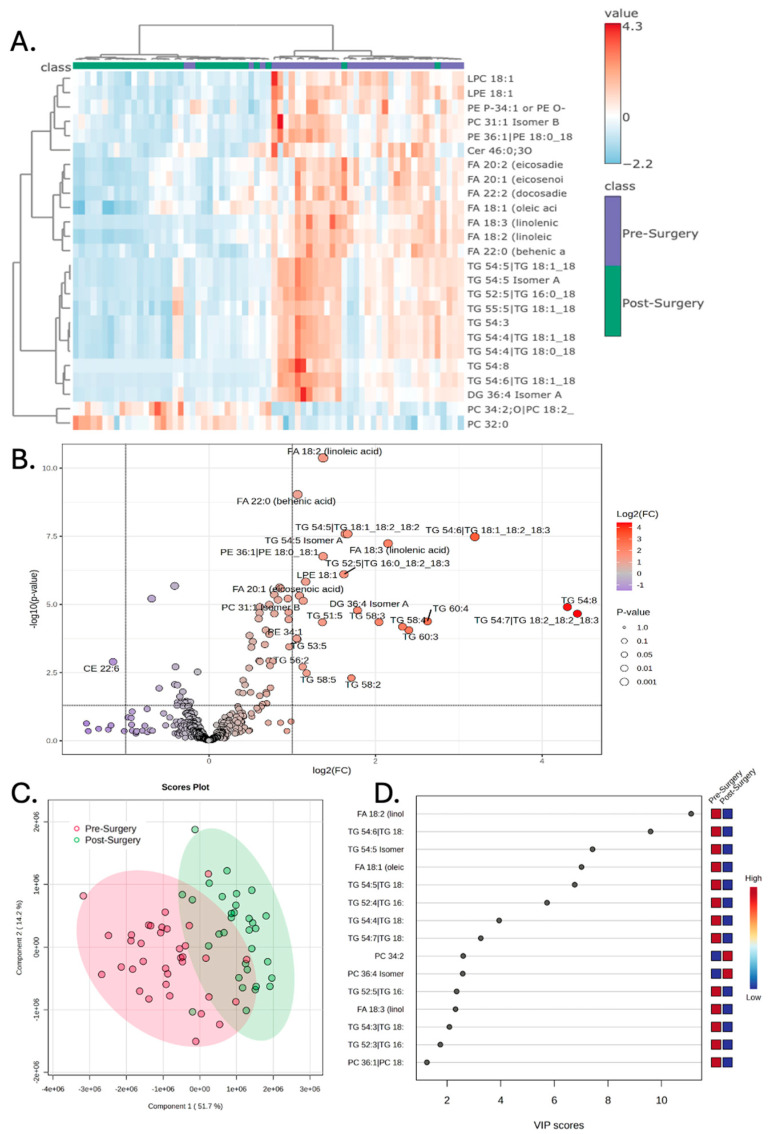
Peripheral lipidomics of patients prior to surgery demonstrates increased abundance of lipids that are decreased following surgery. (**A**) Hierarchical clustering of patient samples based on the top 25 lipid metabolites with the highest F-statistic values from a one-way ANOVA. These lipids represent the greatest variance across the treatment stages and demonstrate that pre-surgery samples have a distinct lipidomic profile compared to post-surgery. (**B**) Volcano plot demonstrating numerous lipids that have increased compound abundance prior to surgery; significance drawn at fold-change greater than two and *p*-values of less than 0.05 after correcting for a false discovery rate of less than 0.05. (**C**) Partial least square discriminant analysis (PLS-DA) demonstrating that the pre-surgery samples separate from post-surgery samples when the first and second principal components are examined. (**D**) Variable importance in projection (VIP) scores of the 15 lipids that hold the largest contribution to the dataset’s variance.

**Figure 3 metabolites-15-00673-f003:**
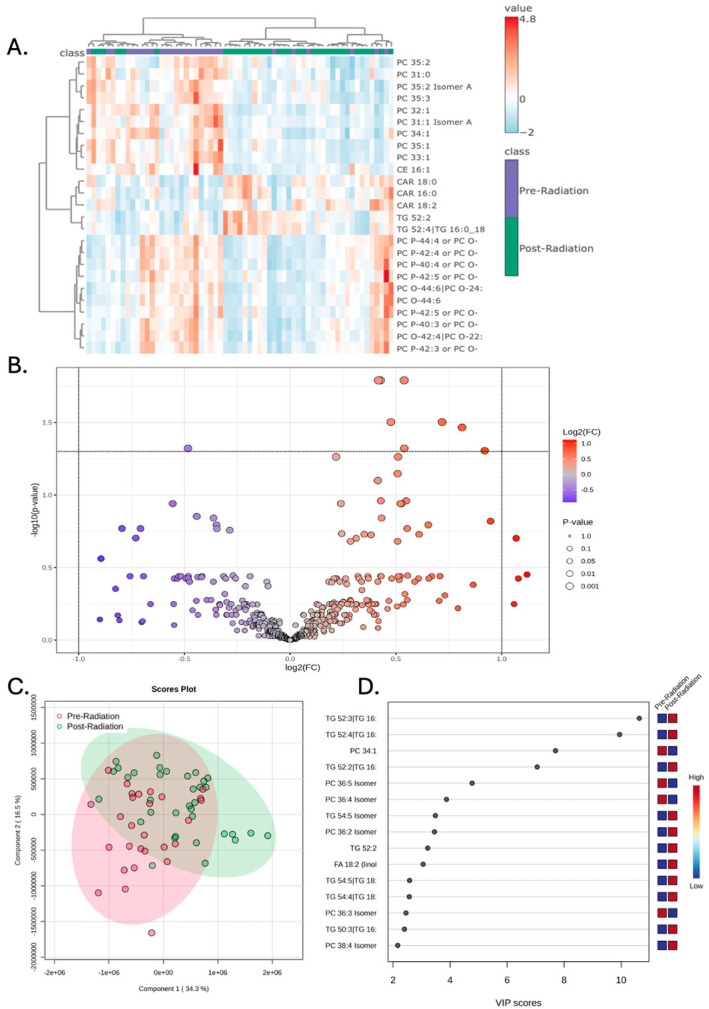
Peripheral lipidomics of patients with glioblastoma changes with radiation to a lesser extent compared to the change observed with surgery. (**A**) Hierarchical clustering of patient samples based on the 25 lipid metabolites with the highest F-statistic values from a one-way ANOVA. These lipids represent the greatest variance across the treatment stages and demonstrate that pre-radiation samples have a distinct lipidomic profile compared to post-radiation samples. (**B**) Volcano plot demonstrates increased compound abundance of three lipids and decreased compound abundance of one lipid prior to radiation; significance drawn at fold-change greater than two and *p*-value less than 0.05. (**C**) Partial least square discriminant analysis (PLS-DA) demonstrates that the pre-radiation samples have poor separation from post-radiation samples when the first and second principal components are examined. (**D**) Variable importance in projection (VIP) scores of the 15 lipids that hold the largest contribution to the dataset’s variance.

**Figure 4 metabolites-15-00673-f004:**
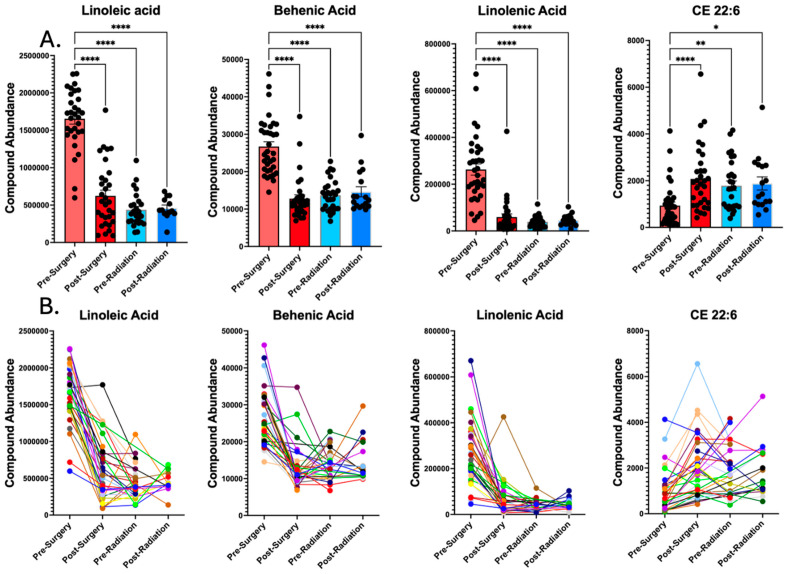
Plasma lipid abundances are reduced following surgery and remain low at later treatment stages. (**A**) Compound abundances of linoleic acid, behenic acid, and linolenic acid are decreased following surgery and remain low during the pre-radiation and post-radiation treatment stages; however, cholesteryl ester (CE) 22:6 increases following surgery and remains elevated at later treatment stages. (**B**) Individual abundance profiles of the same lipid data as in section A are depicted for visualization of intra-individual changes in the lipids across treatment stages, with each individual profile designated by a different line color. Statistical analysis involved repeated measures analysis of variance (ANOVA) with Tukey post-hoc test for multiple pairwise comparisons, with statistical significance drawn at standard *p* value ranges, specifically * *p* < 0.05, ** *p* < 0.01, **** *p* < 0.0001.

**Figure 5 metabolites-15-00673-f005:**
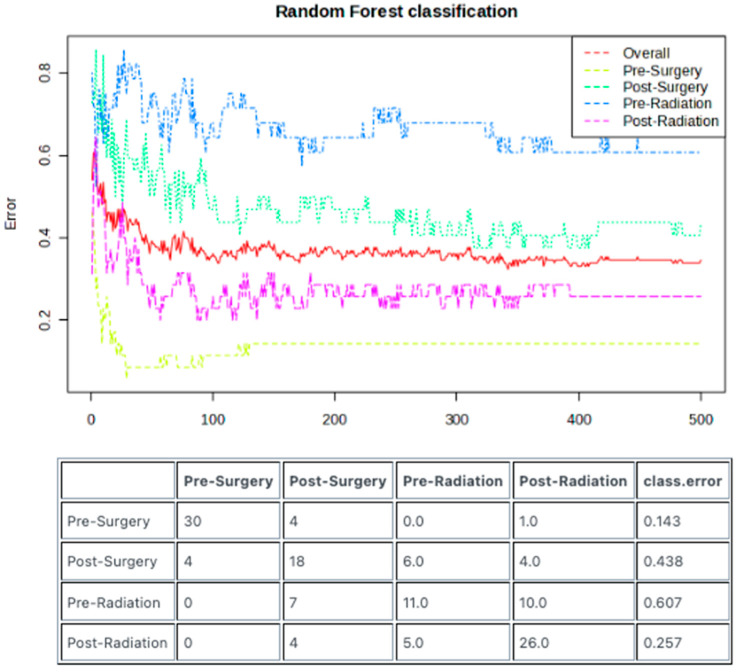
Machine learning algorithm can distinguish between the lipidomic signatures of different treatment stages. Random forest model utilizing 500 trees can identify differences in the lipidomic signatures of each treatment stage. Out-of-bag error demonstrates low predictive error (high accuracy) of 0.143 for samples from the pre-surgery stage and 0.257 for samples from the post-radiation stage. Post-surgery and pre-radiation had high error rates of 0.438 and 0.607, respectively.

**Table 1 metabolites-15-00673-t001:** Demographics of patients, including sex, ethnicity, body mass index at diagnosis, and treatment stage. “X” indicates that a sample was collected at the listed time point for each patient.

Patient ID	Sex	Ethnicity	Diagnosis Age (years)	BMI at Diagnosis	Pre-Surgery	Post-Surgery	Pre-Radiation	Post-Radiation
1	M	White	60	40	X	X	X	X
2	M	White	72	30	X	X	X	
3	M	Hispanic	43	28	X	X		X
4	M	Asian	49	57	X	X		X
5	F	White	78	23	X	X		
6	M	Hispanic	65	22	X	X		X
7	M	White	72	41	X	X	X	
8	M	White	80	24	X	X	X	X
9	F	White	61	27	X	X	X	
10	F	White	69	25	X	X	X	
11	M	Indian	60	27	X		X	X
12	F	White	61	25	X	X	X	
13	F	White	52	27	X	X		
14	M	White	62	30	X	X	X	
15	M	White	69	31	X	X	X	X
16	M	White	67	44	X	X		
17	F	White	82	28	X	X	X	
18	F	White	55	29	X	X		
19	M	African American	47	37	X	X	X	X
20	M	White	63	30	X	X	X	X
21	F	White	86	27	X	X	X	
22	F	White	64	31	X	X	X	X
23	M	White	56	22	X	X	X	X
24	F	White	69	26	X	X	X	X
25	F	NA	69	27	X	X	X	
26	M	White	64	36	X	X	X	X
27	M	White	68	28	X		X	X
28	M	White	69	28	X	X	X	X
29	F	White	58	27	X		X	X
30	F	white	66	27	X	X	X	
31	M	White	55	28	X	X	X	X
32	F	White	60	20	X	X	X	
33	M	White	58	28	X	X	X	
34	M	White	53	30	X	X	X	
35	M	White	58	26	X	X	X	
36	M	White	76	35	X			

**Table 2 metabolites-15-00673-t002:** List of the top 25 lipids with the most significant change in abundance following surgery. *p* values are corrected for a false discovery rate of less than 0.05. All lipids had higher abundances prior to surgery, except for CE 22:6, which had lower abundance. Abbreviations include triglycerides (TGs), phosphatidylethanolamines (PEs), phosphatidylcholines (PCs), diglycerides (DGs), lysophosphatidylethanolamines (LPEs), and cholesteryl docosahexaenoate (CE).

Lipid Name	Fold-Change	*p*-Value
Linoleic acid	2.58	4.21 × 10^−11^
Behenic acid	2.09	9.3 × 10^−10^
TG 54:5 Isomer A	3.11	2.58 × 10^−8^
TG 54:5	3.17	2.58 × 10^−8^
TG 54:6	9.14	3.33 × 10^−8^
Linolenic acid	4.44	5.83 × 10^−8^
PE 36:1	2.58	1.73 × 10^−7^
TG 52:5	3.07	7.84 × 10^−7^
LPE 18:1	2.24	1.46 × 10^−6^
Eicosenoic acid	2.12	4.79 × 10^−6^
PC 31:1 Isomer B	2.19	7.38 × 10^−6^
TG 54:8	19.8	1.25 × 10^−5^
DG 36:4 Isomer A	3.44	1.66 × 10^−5^
TG 54:7|TG 18:2_18:2_18:3	21.5	2.19 × 10^−5^
TG 60:4	6.16	4.17 × 10^−5^
TG 58:3	4.11	4.45 × 10^−5^
TG 51:5	2.57	4.51 × 10^−5^
TG 58:4	5.01	6.60 × 10^−5^
TG 60:3	5.28	8.77 × 10^−5^
PE 34:1	2.07	1.78 × 10^−4^
TG 53:5	2.08	1.90 × 10^−4^
CE 22:6	0.45	1.26 × 10^−3^
TG 56:2	2.18	1.94 × 10^−3^
TG 58:5	2.25	3.31 × 10^−3^
TG 58:2	3.27	4.99 × 10^−3^

## Data Availability

All data can be made available upon request.

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
