# Peer review of "Longitudinal Plasma Lipidomics Reveals Distinct Signatures Following Surgery in Patients with Glioblastoma"

_metabolites, 2025, doi:10.3390/metabo15100673_

Round 1
Reviewer 1 Report
Comments and Suggestions for Authors
I praise the authors for undertaking research in glioblastoma by engaging a lipidomics approach, both are important and challenging fields. Nevertheless, the current manuscript does not yet meet the standards expected for scientific merit and clarity. My major concerns are as follows:
Overall
- Title: The duration of the study period on the same subject is not sufficient to be claimed as a longitudinal study.
- The data presented and discussed do not adequately support the stated aim of biomarker discovery for diagnosis or disease monitoring. While the authors briefly acknowledged study limitations, the current findings appear more indicative of treatment regimen–related lipid alterations rather than diagnostic or prognostic markers.
Methods:
- Section 2.1: The timing of radiotherapy initiation relative to surgery is unclear. It is inconsistently referred to as concurrent (line 186) and as occurring in later treatment stages (line 208). Please clarify the timeline.
- The confounding factors such as the patient’s nutritional status and medical conditions cannot be ruled out after surgery. These may significantly affect metabolic rate and confound the observed findings.
- More details should be added in the section 2.2: volume of plasma, extraction solvent, and resuspension buffer used; incubation time, centrifuge speed, drying condition; instrument model, column brand and dimension, mobile phases used, gradient, injection volume, and acquisition parameters. Specify the curated lipid library (line 109) used, whether from in-house or public databases. Provide the QCs distribution before and after batch effect correction as supplementary materials.
- Statistics: Specify the type of ANOVA used for these paired samples.
Results:
- Figures 1-3A: Clarify the criteria used to define the top 25 lipids. Provide the permutation test and CV model results for PLS-DA as supplementary materials.
- The description of the PLS-DA score plots (line 144, 161, and 192) is subjective. For example, the claim of "clear" separation in Figure 1C is not visually convincing.
- The respective figure 1-4 represents the same data in various formats, without offering additional insights. Consider consolidating them or presenting them more effectively.
- The result description emphasizes on the pre-surgery group by comparing to post-surgery group, what can the “increased” level in pre-surgery group imply. Without healthy control, it is not sure whether the “increased” levels are truly tumor-related.
- Please check whether p or FDR values were used. They are not consistent in Table, Figures, and text.
- The lipids reported in Figure 4 are very much related to dietary sources. This weakens their interpretation as biomarkers.
- What specific lipids were identified by the RF model? How do these aid in diagnosis, as claimed (line 227)? Why is it necessary to differentiate lipidomic profiles across treatment stages?
Discussion & Conclusions
- The discussion section is overly simplified and does not critically address how the findings meet the study objectives.
- This manuscript would be significantly strengthened by correlating circulating lipids with clinical parameters such as tumor size, tumor biology and patient outcome.
Minor:
- Consider displaying full lipid names in the heatmaps and VIP score plots. The current truncated names in this figures are less informative to readers.
Reviewer 2 Report
Comments and Suggestions for Authors
This study is focused on determining the changes in the plasma lipid profile of IDH-glioblastoma patients at four stages of the disease progression, namely pre-surgery, post-surgery, pre-radiation and post-radiation. The number of patients was sequentially reduced. Liquid chromatography coupled to orbitrap mass spectrometry with positive ionization was used for the untargeted lipidomic analysis. Comparison of the lipid profile at the different stages was done by means of Random Forest statistical modeling.
The experimental set-up, as reported in the manuscript, is well conducted and results are of interest as arising from a prospective study. Limitations of the study are clearly stated by the authors in the manuscript.
Before acceptation, the issues that follow need to be conveniently addressed:
1) the section 2.2. Sample Preparation and Lipidomics Analysis needs improvement: i) was there one extraction step only, that is, was the water phase extracted with MTBE once only?; ii) which solvents were used in the binary mobile phase? and which was the gradient, flow, ...?; iii) peak height: chromatographic or the mass spectrum?; iv) explain a bit more how the SERRF methodology was applied.
2) Even though there not seem to be significant changes, the time interval between surgery and radiation might be relevant to determine lipid variations between post-surgery and pre-radiation. Otherwise, the comparison between pre-surgery and post-radiation might be of interest rather than pre- and post-radiation.
3) It seems that a high number of triacylglycerols (TGs) were shown to have their content enhanced at the pre-surgery stage. This fact could be related to the supply of fatty acids for the enhanced energetic metabolism of the cancer cells in addition to new cell membrane build-up. However, little is discussed on this concern in the manuscript.
4) Text in lines 209-211 has nothing to do with that shown in figure 4!! Furthermore, figure 4B does not allow to follow evolution, this being an interesting result, and hence needs improvement.
5) Lines 270-271: CE 22:6 is not a fatty acid. An anex with abbreviations would be welcome
Round 2
Reviewer 1 Report
Comments and Suggestions for Authors
I appreciate the authors’ efforts in addressing the comments. Regrettably, I am not convinced that the revisions sufficiently resolve the concerns raised regarding the study design, the findings and their interpretation and discussion. Therefore, I am unable to recommend acceptance of the manuscript in its current form.
Reviewer 2 Report
Comments and Suggestions for Authors
I recommend two issues to be improved:
1) Line 120: indicate the time the gradient was running, that is the duration of the chromatographic run.
2) As previously indicated, figure 4B is poorly informative since it is not possible to follow the trends. Improvement of it would be acknowledged.
